# Enhancing Airside Monitoring: Multi-view Approach for Accurate Aircraft Distance-To-Touchdown Estimation in Digital Towers

## Abstract

A digital tower, a cost-effective alternative to physical air traffic control towers, is expected to provide video-sensor-based surveillance, which is particularly advantageous for small airports. To fully realize this potential, advanced computer vision algorithms play a crucial role in effective airside monitoring. While current research primarily focuses on tracking aircraft on airport surfaces, an equally vital aspect is the real-time observation of approaching aircraft on the runway. This capability holds a pivotal role in augmenting both airport and runway operations. In this context, the study introduces a real-time deep learning approach to accurately estimate the distance-to-touchdown of approaching aircraft, covering distances of up to 10 nautical miles. The approach overcomes the limitations of monoscopic and stereoscopic methods by utilizing multi-view video feeds from digital towers. It integrates Yolov7, an advanced real-time object detection model, with auxiliary regression auto-calibration, enabling real-time tracking and feature extraction from diverse camera viewpoints. Subsequently, an ensemble approach utilizing an LSTM model is proposed to combine input vectors, resulting in precise distance estimation. Notably, this approach is designed for easy adaptation to various camera system configurations within digital towers. The model's effectiveness is assessed using simulated and real video data from Singapore Changi Airport, demonstrating stability across scenarios with low predictive errors (Mean Absolute Percentage Error = 0.18%) up to 10 nautical miles under visual meteorological conditions. These capabilities within a digital tower environment can significantly enhance the controller's ability to manage runway sequencing and final approach spacing, ultimately leading to remarkable airport efficiency and safety improvements.

## 1 Introduction

Digital towers have emerged as a promising solution for replacing physical towers in small and medium-sized airports, and they are also being incorporated into the development of larger airports as digital twins in conjunction with their physical counterparts. These digital towers rely on video data captured by an array of cameras, which are expected to provide surveillance capabilities for airports without expensive radar systems or complement existing surveillance systems in large airports and enhance their performance in terms of safety and efficiency. Numerous studies have demonstrated the advantages of utilizing multi-sensor data to manage airport operations, enabling better situational awareness of air traffic movements on the ground and during the final approach phase Sturdivant & Chong (2017). Furthermore, an intriguing investigation Papenfuss & Friedrich (2016) has indicated that the augmentation of airport situational information directly on screens has the potential to reduce controller workload and improve their performance. However, the integration of digital tower systems and video feeds with the existing surveillance systems poses significant challenges, impeding their effective exchange of information. Consequently, the benefits of a digitized environment in airport operations remain constrained. To address this limitation, a potential approach is the development of a suite of computer vision algorithms capable of leveraging video streams to derive decision-making information. In recent years, the academic literature has witnessed a growing body of research on applying computer vision techniques in the airport envi-

ronment. These studies have explored various aspects, including aircraft tracking, airport surface surveillance Zhang et al. (2020); Van Phat et al. (2021); Li et al. (2022); Zhang et al. (2022), monitoring airport apron activities and aircraft turnaround processes Lyu et al. (2022); Thai et al. (2022), as well as enhancing airport safety through debris and drone detection Qunyu et al. (2009); Noroozi & Shah (2023); Thai et al. (2019). Notably, while several investigations have delved into these areas, there exists a notable research gap in terms of estimating the distance-to-touchdown (DTD), a critical parameter essential for final approach spacing and departure sequencing. Integrating such estimation capability within a Digital Tower environment holds significant potential for enhancing runway controllers' sequencing and final approach spacing abilities.

In the specific task of estimating the distance of moving objects, machine learning techniques have garnered considerable attention. Two primary approaches can be discerned for computer vision-based distance estimation: stereoscopic and monoscopic view methods. Stereoscopic approaches involve using two cameras to capture video data, with distance estimation relying on calculating the disparity between objects (or pixels) observed on the two camera screens. However, the calibration and proper alignment of these cameras pose notable challenges Strbac et al. (2020). Inaccuracies during the calibration or pixel matching processes can introduce substantial errors in the distance estimation, particularly for objects located at greater distances. Several studies have explored this research direction, particularly employing street view datasets. For instance, a study proposed a rapid and accurate algorithm leveraging stereo data to recover dense depth information from stereo video, assuming a static scene Yamaguchi et al. (2014). Additionally, with the advent of high-performance object detection algorithms, such as You Only Look Once (YOLO Redmon et al. (2016)), subsequent research has begun utilizing object detection as an intermediate step for distance or depth estimation. For example, a recent study presented a distance estimation solution based on the YOLO deep neural network and principles of stereoscopy Strbac et al. (2020). However, it is important to note that these approaches have primarily demonstrated satisfactory results for static objects in close proximity. In contrast, monoscopic approaches employ a single camera for distance estimation. As direct recovery of depth information is not feasible from a single camera, it typically relies on object detection techniques to identify objects using bounding boxes. The detected bounding box, the object's size, and referenced markers are then utilized to estimate the object's distance from the camera. For instance, a straightforward approach utilizing the You Only Look Once (YOLO) algorithm is presented in Abdul et al. (2019). This approach has been evaluated in various environments and demonstrates promising results with different monocular cameras, achieving a vision range of up to 1000m. Another notable work in this vein is the DepthNet framework Masoumian et al. (2021), which encompasses a more intricate deep learning architecture comprising two networks for depth estimation and object detection using a single image. However, training DepthNet poses significant challenges in achieving high accuracy, particularly for long-distance estimation. One of the primary hurdles lies in accurately detecting the bounding box of small objects and identifying referenced markers within the scene, which proves to be particularly demanding in the context of approaching aircraft. Moreover, a recent noteworthy study Tang et al. (2018) explores an inter-camera (multi-views) approach focusing on vehicle tracking and speed estimation. This research highlights the potential benefits of leveraging multiple cameras to enhance distance estimation performance.

In summary, while advanced deep learning algorithms have exhibited notable achievements in street view and indoor datasets, a universal solution that can effectively address all challenges remains elusive, particularly in featureless blue skies scenarios. Furthermore, achieving high accuracy in estimating the distance (up to 10NM) of small moving objects poses a significant challenge, necessitating the exploration and proposal of novel approaches to meet the required performance criteria. Building upon the advancements presented in the aforementioned studies, this work presents a novel multi-view vision-based deep learning approach for estimating the Distance-to-touchdown (DTD) of approaching aircraft. The proposed approach leverages multi-camera video feeds to combine the strengths of existing methodologies. This paper aims to make several contributions to the field, as follows.

1. The model's architecture incorporates calibration and sequential layers to ensure stable performance, accounting for factors such as noisy input or errors in object detection algorithms. This design choice enables the model to handle the stochastic nature of input video feeds effectively.

2. In the evaluation phase, simulated and real video data from Changi Airport are utilized to assess the model's performance. Notably, the model demonstrates remarkable capability in challenging scenarios, including low visibility, stormy weather, and low light conditions.

3. Leveraging a pre-trained model for object detection and employing auto-segmentation techniques significantly reduce data requirements and training time while achieving high accuracy in estimating the distance to touchdown (DTD) up to 10 NM.

4. To tackle potential changes in configurations of the camera system, a calibration network trained with an auxiliary regression head is proposed, further enhancing the model's adaptability and robustness.

## 2 THE PROPOSED APPROACH FOR DISTANT-TO-TOUCHDOWN ESTIMATION

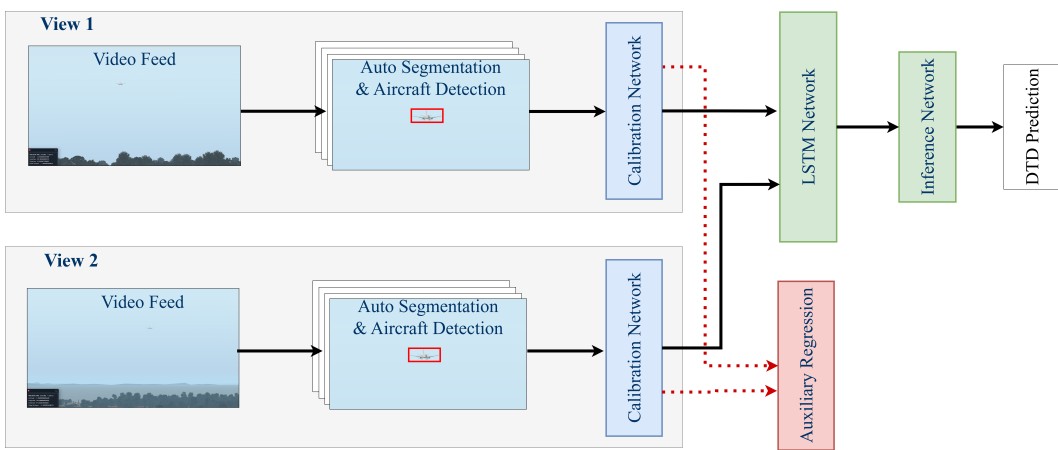

Figure 1: The concept diagram of the proposed approach for Distance-to-Touchdown (DTD) Prediction.

The concept diagram of the proposed approach is presented in Figure 1. The model has two main parts: the final feature vector extraction from each camera view and the ensemble component for estimating the DTD. It is designed to tackle operational challenges such as varying numbers of camera views for runway operations at different airports and potential changes in the camera's configuration, such as angle and zoom. In those cases, most end-to-end computer vision models must be retrained or fine-tuned with the newly collected data to maintain performance.

First, video feeds from 2 camera views are utilized as the model's inputs. To obtain the final feature vector for each video feed, the sequence of images is input into the auto-segmentation module for localizing the potential aircraft position using an aircraft detection model and cropping the redundant video frames' areas. This step helps remove unnecessary information in the image and keep the size of the approaching aircraft, which, by design, is far away and very small. Then, the bounding boxes of the detected aircraft are input into the fully-connected layers, called calibration networks, for extracting the final feature vectors. All the calibration networks are connected to an auxiliary regression head for training their parameters. This step is necessary for adjusting inputs from different camera views without requiring manual system calibration. The feature vectors are combined using an LSTM model Hochreiter & Schmidhuber (1997) and fully-connected layers for distance prediction. The sequential model combines the multi-view camera inputs to provide the system's stability in case of aircraft detection errors in each video feed. The following sections will discuss the model's architecture, implementation, and training.

## 3 DATA COLLECTION

To train the network effectively, a relatively large dataset featuring multiple views of aircraft on their final approach is required. The required videos must be of high resolution to enable identification of the aircraft at further distances. Additionally, the dataset must also contain the positions of the

aircraft, allowing for the calculation of the distance-to-touchdown. Due to the specifications, high-fidelity simulated data of approaching aircraft is generated for training and validating the proposed approach.

The simulation must feature highly accurate aircraft models and support scripting to reliably place the camera position and angle. It must also enable data exchange with external systems for saving the recorded video feed and aircraft position data. To fulfill these requirements, X-Plane 11 by Laminar Research is selected. It provides various weather conditions with different visibility and lighting scenarios, enabling the generation of non-ideal scenarios to test the model's performance. A Python-based tool based on the XPPython3 Buckner (2017) plugin is developed to expedite the data collection. This tool automatically captures video feeds from designated camera positions and records aircraft locations. Table 1 presents the values of the controlled parameters in our data collection process. The dataset contains videos for 100 scenarios with the corresponding 4D aircraft trajectories, using aircraft model B737 and Changi Airport 3D model (refer to Figure 2). Due to its common usage, the B737 aircraft model was selected as the sole model for data generation in this research. The lighting (time of the day) and weather conditions are adjusted to cover scenarios with different visibility, while the initial randomized location is utilized to create the variation in aircraft position during landing. Noting that, for each scenario, the trajectory's length is around 10NM while the corresponding videos are around 4.0 minutes and recorded from 2 different camera views. In the simulation, those 2 camera viewpoints are placed on both sides of runway 20C and near its Instrument Landing System (ILS). Finally, the visibility in the collected dataset is mostly more than 10NM for training and testing, except in the case of "foggy", where it is designed to reduce the visibility to 5NM.

| Simulation Parameter | Selected Values |
|---|---|
| Airport | Singapore Changi Airport |
| Runway | 02L |
| Aircraft Model | B737 |
| Time of the day (5) | 6:00, 8:00, 12:00, 17:00, 18:00 |
| Weather condition (4) | Clear, Cloudy, Stormy, Foggy |
| Initial positions | Randomized with DTD = 10NM |
| Number of Camera Views | 2 |
| Camera Resolution | 1920 x 1280 |
| Frame rate | 30 FPS |

Table 1: The selected values of simulation parameters for data generation using the X-Plane 11 flight simulator.

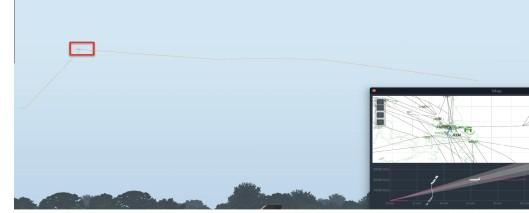

Figure 2: An example of an aircraft at 6NM with its projected trajectory and the corresponding vertical profile.

## 4 EXPERIMENTAL SETTING

This work trains and tests the proposed model with two camera views. Using the simulated video data, 70% of the data (70 scenarios) is used for training, and the remaining data (30 scenarios) is used for testing. Data samples with at least one detected aircraft are used to facilitate the training and evaluating. In total, the training data includes $\approx 496k$ data samples. Regarding the learning algorithm, the YOLOv7 Wang et al. (2022) is adopted as the aircraft detector over each camera view. To combine the extracted information from all the cameras, a stacked LSTM model is developed. More importantly, to reduce the inference time of the proposed end-to-end model, we adopt the TensortRT engine Vanholder (2016) for video processing and aircraft detection steps. As experimented, the processing speed increased up to 300% compared to the model without the TensortRT engine. Moreover, experiments have been conducted for different network architectures as a result, the selected architecture for the calibration network is Linear([256,1]), and for the predictive model is [LSTM([256, 256],2), Linear([256, 1])].

The metrics used to analyze the model performance are Percentage Error (PE) and Mean Absolute Percentage Error (MAPE). The Percentage Error (PE) reports the difference between the actual distance ($A_i$) and the predicted distance ($P_i$) for each predicted distance instance ($i$). It is useful for observing the changes in accuracy over variations of the aircraft distance. On the other hand, the average model performance is assessed using Mean Absolute Percentage Error (MAPE), which is generally suitable for model comparison given the dataset with size $n$. Finally, this project is implemented using PyTorch 1.13 with Python 3.10, and all the training is done on a single RTX 3080. The total training time for the model convergence is $\approx 2$ hours.

In this work, a monoscopic model is developed for benchmarking by estimating distance as a function of aircraft size and bounding box size Abdul et al. (2019). Two independent models are developed for each view, and their performances are average for comparison.

## 4.1 LEARNING ALGORITHM

### 4.1.1 THE ADOPTED AIRCRAFT DETECTOR

The aircraft detection aims to localize the approaching aircraft in the video frame and determine its corresponding bounding box (refer to Figure 3). This research selects a pre-trained YOLOv7 object detection algorithm to reduce the necessary training data. To handle the difference in size between model input and raw video frames, while the conventional solution to this issue is to resize the image to the desired resolution, this is not feasible for approaching aircraft since they may only be a few pixels wide in the frame. Resizing the image would make the aircraft too small, leading to more aircraft detection and tracking challenges. To avoid this issue, an Auto Segmentation algorithm is developed to identify the region of interest in the high-resolution image. Firstly, the high-resolution video frame (1920x1280x3) is split into six non-overlapping image tiles (640x640x3) for aircraft detection. Once an aircraft is detected in any image, a corresponding image (640x640x3) with that aircraft at the center is extracted and passed through the object detection algorithm again for further analysis. This second pass for aircraft detection addresses the issue where the aircraft crosses the tiles' borders and is not detected properly. The correct bounding box can be obtained by running the detection algorithm on the cropped image. The bounding box information, e.g., center location (X and Y) and size (W and H) is calculated corresponding to the coordinates in the original frame. The main purpose of the splitting or segmentation is to keep the sufficient size of the far-away aircraft (up to 10 NM) in the image. With more than one aircraft in the frame, the same number of final images can be generated and follow the same process independently.

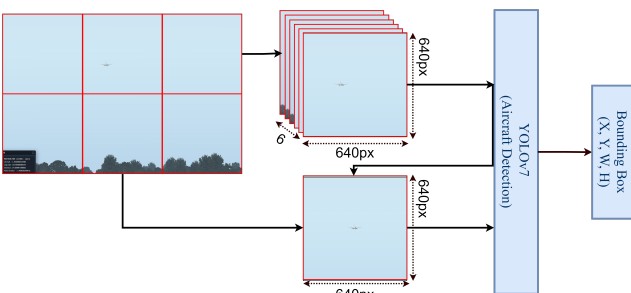

Figure 3: The Auto Segmentation process's concept diagram to localize airplanes in the video frame and extract corresponding bounding boxes.

The proposed Auto Segmentation algorithm is tested on a simulated dataset and compared to simply scaling the original frame to 640x640x3. The results are presented in Table 2, which shows a significant improvement in performance using the proposed algorithm. The maximum range refers to the distance at which the first detections of an aircraft occur, while the effective range is the distance at which consistent detections are observed, with a success rate of over 99%. The Auto Segmentation algorithm achieves a much wider detection range and more stable detections.

| Algorithm | Maximum Range (NM) | Effective Rang (NM) |
| --- | --- | --- |
| Auto Segmentation | 9.95 | 9.74 |
| Conventional Scaling | 7.00 | 4.96 |

Table 2: The maximum and effective range of the aircraft detection algorithm with and without the proposed Auto Segmentation algorithm.

This research aims to provide real-time DTD prediction, so computational time is crucial. The image-splitting algorithm has been optimized to achieve this goal, and the data flow has been streamlined. Additionally, the Yolov7 object detection inference time has been minimized using the TensortRT engine[43] and batched inference. The result of these optimizations is an average

inference time of 28 milliseconds. It is a significant improvement compared to the unoptimized model, which took approximately 100 milliseconds to infer an image used in this research. Finally, this approach emphasizes using state-of-the-art object detectors rather than training or fine-tuning them specifically for aircraft detection. As the object detector model can be exchanged with any other high-performance aircraft detection model, the proposed approach is expected to maintain high performance with minimal modifications.

### 4.1.2 AN ADAPTIVE ALGORITHM FOR TRAINING CALIBRATION NETWORKS

One of the computer vision model's typical limitations is the camera configuration sensitivity, especially when working with multiple cameras. Since the central idea of our approach is the ensemble of multi-view videos for stabilizing the model performance, it must be able to handle the change in the number or the configuration of cameras without the need to retrain the whole model. Therefore, a calibration network is proposed for each camera. It is fully-connected layers designed to construct the feature vectors from detected bounding boxes considering the differences in camera configuration. The auxiliary regression with a reversed network structure is connected to calibration networks for training using DTD values as the target and MAPE as performance metrics.

As the values and qualities of inputs from each camera can have a different impact during training, joint training for multiple cameras may result in a significant gap in the accuracy of the predicted DTD between each camera. This gap may be caused by the values of one camera being more influential than the other. To address this issue and further enhance the quality of the Calibration Network, an adaptive algorithm has been developed (Algorithm 1) for the training. The convergence curve of the training process can be observed in Figure 4. As the network of view 2 converges much faster, and the gap is significant, it is frozen to focus the training on the other network until the performance gap is less than a defined threshold. This process is repeated until both networks are converged with the desired performance gap. The outputs of the calibration networks are the feature vectors used as the input for the distance estimation model. Moreover, to ensure that features across all calibration networks are similar or perspective-independent, a regularization is added to the loss function during the training of the auxiliary head (refer to Equation 1.

$$Loss_i = MSELoss(d_{Pred_i}, d_{Actual}) + MSELoss(d_{Pred_i}, d_{Pred_{1-i}}) \tag{1}$$

This approach can improve the accuracy of the predicted DTD, and the computational costs associated with retraining or fine-tuning the entire architecture can be reduced. Therefore, when a new or adjusted camera input is added to the system, only its calibration network must be trained with the frozen auxiliary regression head while the whole system remains unchanged.

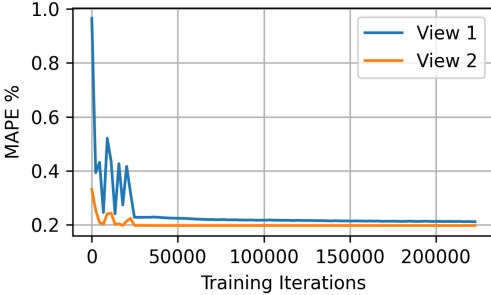

Figure 4: The convergence curve of the proposed adaptive algorithm for training the calibration layers of two camera views.

### 4.1.3 DTD PREDICTIVE NETWORK

The Distance Estimator component serves as the final stage in the proposed ensemble architecture and is responsible for predicting the final DTD of the aircraft. This component combines the feature vectors extracted by the Calibration Networks for all camera views, enhancing the stability and accuracy of the predicted distances. The Distance Estimator comprises two sub-components: an LSTM and a fully-connected network.

---

**Algorithm 1** The Proposed Adaptive Algorithm.

---

**Input**: Images from $N\_views$
**Parameter**: $N\_views = 2$, $max\_epochs = 225K$, Min $= 1500$, $\Delta_G = 0.01$, $\Delta_L = 0.01$
**Output**: $calnets$

1:   $calnets \leftarrow$ a list of $N\_views$ calibration networks
2:   $loss \leftarrow$ an empty list with size $N\_views$
3:   $optimizing\_views \leftarrow$ range($N\_views$)
4:   $loss\_log \leftarrow []$
5:   $steps \leftarrow 0$
6:   **for** $e \leftarrow 1$ to $max\_epochs$ **do**
7:     **for** $vid \in optimizing\_views$ **do**
8:       $loss[vid], calnets[vid] \leftarrow$ Optimize($calnets[vid]$)
9:     **end for**
10:   $loss\_log$.append($loss$)
11:   $steps \leftarrow steps + 1$
12:   **if** ($steps >$Min)&(std($loss\_log[-3, :]$) $< \Delta_L$) **then**
13:     $loss\_gap \leftarrow loss - $min($loss$)
14:     $optimizing\_views \leftarrow$ Index($loss\_gap > \Delta_G$)
15:     **if** $optimizing\_views$ is empty **then**
16:       $optimizing\_views \leftarrow$ range($N\_views$)
17:     **else**
18:       $steps \leftarrow 1$
19:     **end if**
20:   **end if**
21: **end for**
22: **return** $calnets$

---

The LSTM network in the Distance Estimator is designed to fuse feature vectors from all cameras into a unified representation. This fusion of information from multiple calibrated camera views enhances the stability and accuracy of the predicted distances. Moreover, the sequential nature of LSTM architecture allows for handling varying numbers of cameras or maintaining the model accuracy by skipping miss-detection in any of the camera inputs. Subsequently, the fully-connected network is responsible for predicting the final DTD. This network shares similarities with the Auxiliary Regression Head, but it operates on the unified feature vectors generated by the LSTM network. In this research, the Distance Estimator network is trained by randomly dropping feature vectors from one of the cameras for each mini-batch. This approach is adopted to ensure that the final model can provide sensible predictions even if there are miss-detections in any video input. Overall, this approach allows for integrating information from all available camera views, improving the overall accuracy and stability of the predicted distances.

## 5   RESULTS AND DISCUSSION

Out of 476k data points, there are 71k cases (15%) where miss-detections in either camera view are observed. Additionally, this research found that the position and size of the aircraft's bounding box have a strong correlation with its DTD. The position of the bounding box (refer to Figure 2) is a better predictor for DTD. Moreover, the experimental results are reported and discussed in the rest of this session to assess the proposed approach's advantages.

The proposed approach (multi-camera) and monoscopic model performance are presented in Figure 5. In general, the proposed approach achieves smaller errors and more stable results. The MAPE (0.18%) reduces by 67% compared to the MAPE $= 0.58\%$ of the monoscopic model. Up to 5NM, the proposed approach achieves high performance with median errors close to zero and a small standard deviation. On average, the performance of both models is comparable from 8NM to 10NM, which is expected due to the limitation of the pre-train detection model for small flying aircraft. However, around 7NM, the error of the proposed approach becomes higher than the monoscopic model. As defined in Changi Airport's Instrument Approach Chart (AIC), there are two Distance Measuring Equipment (DME) points at 4.4NM (also the Final Approach Fix (FAF)) and 7.6NM. The

aircraft would adjust its altitude between those points before crossing the FAF (refer to Figure 2). It can be observed from the video feeds that, during that period, the positions and sizes of bounding boxes were indifferent. Thus, estimating distance based on those detected bounding boxes leads to higher errors (up to $0.75\%$ or 100m) than the other periods. These results indicate that the proposed approach is highly effective in accurately predicting the DTD for approaching aircraft, especially when multiple cameras are utilized.

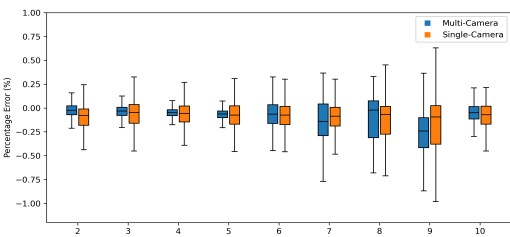

| | | Dual Exclusive | | Combination | |
|---|---|---|---|---|---|
| | | MAPE (%) | SD | MAPE (%) | SD |
| **Weather** | **Clear** | 0.10 | 0.15 | 0.18 | 0.75 |
| | **Cloudy** | 0.22 | 0.68 | 0.3 | 0.91 |
| | **Foggy** | 0.09 | 0.09 | 0.21 | 0.70 |
| | **Stormy** | 0.11 | 0.15 | 0.18 | 0.57 |
| **Time** | **06:00** | 0.13 | 0.27 | 0.25 | 0.79 |
| | **08:00** | 0.14 | 0.18 | 0.23 | 0.66 |
| | **12:00** | 0.18 | 0.52 | 0.25 | 0.75 |
| | **17:00** | 0.16 | 0.18 | 0.23 | 0.62 |
| | **18:00** | 0.13 | 0.14 | 0.19 | 0.55 |

Figure 5: Experiment results for comparison between the proposed model (Multi-Camera) and the benchmarking monoscopic model (Single-Camera) in terms of Percentage Error (%).

Table 3: Impact of weather and lighting conditions on model performance: detected aircraft in both views (Dual Exclusive) and detected aircraft in at least one view (Combination)

Furthermore, the ability of the proposed model to perform accurately in various weather and lighting conditions is crucial for a DTD prediction system to be practical and reliable in real-world scenarios. Therefore, further evaluation of the model's performance under different environmental conditions is crucial as it can affect the quality of camera feeds and, subsequently, the model's accuracy. The following analysis focuses on the performance of the proposed model under four weather conditions and five lighting conditions for two different settings (frames with detected aircraft in both views, called Dual Exclusive, and frames with detected aircraft in at least one view, called Combination), refer to Table 3. For the impact of weather on model performance, the maximum error for the cloudy condition ($MAPE = 0.3\%$) is much higher than the others, while the smallest MAPE values are for clear and stormy scenarios. A potential explanation is that the inconsistency of the sky due to clouds reduces the accuracy of the aircraft bounding box, and the sky with clear or stormy conditions is much more consistent than in cloudy cases. For the impact of lighting conditions on prediction accuracy, the scenarios at 6:00 and 18:00 are considered in low-light conditions, while from 8:00 to 17:00, the light is much more intense. It is interesting to observe a better performance of the model in some low-light conditions. Our further inspections and analyses of those cases suggest that the reflected glare and shadow have affected the accuracy of the bounding box estimation. Moreover, utilizing the dual exclusive setting generally provides much higher performance by reducing more than $30\%$ errors in distance estimation compared to the combination setting. It emphasizes the essential of adopting a high-performance aircraft detection model in the framework.

In summary, the proposed model architecture for aircraft distance estimation using multiple camera views has shown to be effective in addressing operational challenges in digital towers without the need for extensive training. The use of multiple camera views has improved the accuracy and stability of the distance estimation. The model's performance is consistent across different lighting conditions, except for slightly higher errors observed during high-light intensity periods. The weather condition strongly impacts the model's performance, with cloudy conditions affecting the accuracy of the detection and bounding box estimation. Despite these challenges, the proposed approach has demonstrated its ability to handle different environmental conditions extremely well, highlighting its potential for practical use in real-world scenarios.

## 6    Case Study of Changi Airport

The objective of the ensuing experiment is to assess the effectiveness of the proposed approach using real-world data. A dataset with 100 landing trajectories for Runway 02L at Singapore Changi Airport is used for this purpose. The sky is cloudy during the data collection; thus, all the collected videos are under cloudy conditions. Two camera views for each landing trajectory are recorded. View 1, including 2 cameras, is the view from the control tower, and View 2, including 3 cameras, is from the ILS. Besides, the visibility in this dataset is less than 7NM due to the weather conditions.

The raw videos from different camera views are processed and merged to create the final dataset. The example of videos from those views in the final dataset can be observed in Figure 6. Moreover, the aircraft in the video data must also be matched with their respective recorded trajectories from the surveillance system for obtaining the DTD information. One significant limitation pertains to the cameras' slightly different positions and perspectives that comprise the views, resulting in an imperfect final image. This slight misalignment is often tolerable; however, the object detector might falsely detect an object in certain cases. Consequently, the object detector considers these objects as two distinct entities despite their similarity, leading to inaccuracies in predicted bounding boxes and, ultimately, reducing the model's performance.

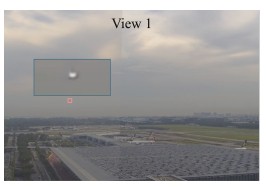 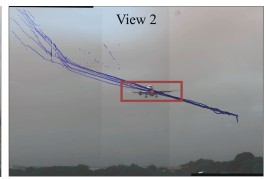 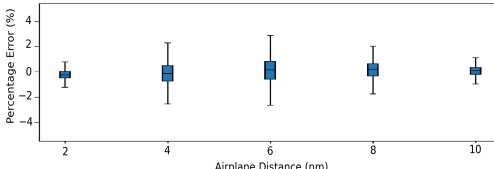

Figure 6: An example of video data from Changi Airport.

Figure 7: The performance of the proposed approach on real Changi Airport data.

The proposed model is trained and evaluated with the above dataset (70/30). The results are consistent with the findings and model performance of the model trained by simulated data. First, View 2 from ILS can capture the landing trajectory, which is necessary for estimating DTD. However, the aircraft is too small in View 1, and all the videos are also under cloudy conditions; thus, the miss-detection rate for videos from View 1 is very high ($> 90\%$). Thus, in most of the data points for training and testing, only data from View 2 is available. Secondly, our model can work with DTD up to 10NM and achieve the expected performance under cloudy conditions with MAPE: $0.33\%$ (or $< 45m$) and std: $0.42\%$. This performance is within the acceptable error for the DTD estimation. Besides, model accuracy over the DTD can be observed in Figure 7, in which the predicted errors (up to $2\%$ or $216m$) around 6NM are observably higher than those at other distances. It aligns with our previous observation and explanation in Session 5.

## 7    CONCLUSION AND FUTURE WORK

In this work, we propose a multi-view vision-based deep learning approach for Distance-to-touchdown (DTD) estimation up to 10NM under various lighting and weather conditions. The approach is designed to provide stable operation and performance with the stochastic numbers of input video feeds due to noisy inputs or miss-detection. The calibration network and auto-segmentation are proposed for tackling the potential differences and changes in the camera system's configuration. The proposed approach can achieve high and stable performance with Changi Airport simulated data (MAPE = 0.18%) for DTD up to 10NM and real data (MAPE = 0.33% under cloudy conditions) for DTD up to 7NM. It also demonstrates a more stable performance than the monoscopic model, which solely relies on the input from one camera view. In this approach, the aircraft' positions along their trajectories are the key features in the DTD estimation. Therefore, the pattern of the landing trajectories captured in the videos is a factor to be considered to ensure the model's performance. Besides, the lighting and weather conditions add many challenges and uncertainties to the video dataset and strongly impact the predictive accuracy. Moreover, a new set of Changi Airport data is being collected with adjusted View 1 for better visibility, covering more lighting and weather conditions. Finally, the auto-calibration step will be updated, and more experiments will be conducted for transfer learning, where the model can be trained by combining the simulated and real data to adapt to a real airport environment.

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
