# OpenReview forum: "Enhancing Airside Monitoring: Multi-view Approach for Accurate Aircraft Distance-To-Touchdown Estimation in Digital Towers"
_ICLR.cc/2024/Conference — Submitted to ICLR 2024_

### Official Review · Reviewer_E31H · 2023-10-24

**Soundness:** 2 fair
**Presentation:** 1 poor
**Contribution:** 2 fair
**Rating:** 3
**Confidence:** 3

**Summary:**

This paper propose a method to estimate the distance for aircraft in digital towers. Distance/depth estimation is an interesting topic in 3D vision.

**Strengths:**

1. Distance estimation is a challenging task.
2. This paper is easy to understand.

**Weaknesses:**

1. The model design is not novel, which has limited technical learning.
2. The dataset is not available. Then it cannot be a part of contribution.
3. Calibaration network is a little strange. Why it can align two views without knowing the related position for the two cameras? If the two camera's position is changed, can this model still work?
4. The number of views in the dataset is only 2.  The statement of "multi-view" is unsoundness. Author should increase the view number.
5. The paper writing should be further improved. Besides, figure in the manuscript should be the vector figure (Most figures are blur).

**Questions:**

Refer to the weakness.

---

> ### Author Response · Authors · 2023-11-22
> **Discussion on the key contributions of the papers based on reviewer comments**
>
> Thank you for your constructive comments and feedback.
>
> 1. Calibration Network, Auxiliary Network, and Their Training:
> The primary contribution of this work lies in proposing a framework that optimally utilizes multiple view camera streams in airport digital towers to estimate the distance to touchdown of landing aircraft. The approach is grounded in ensemble learning, where each view is processed independently, and their information is subsequently transformed into a shared space for ensemble prediction. The proposed framework is designed to provide flexible adaptation for different digital tower configurations. Depending on the available airport data and its corresponding challenges, the model for aircraft detection can be customized to maximize the system's performance.
>
> Besides, the calibration network plays a pivotal role in handling multiple cameras by converting input features from individual views into a shared feature space. Furthermore, an auxiliary network is shared and employed by all calibration networks to train their parameters.
>
> Furthermore, the current choice of the number of views (e.g., 2) aligns with the current operational constraints of digital towers. However, Algorithm 1 is designed to accommodate arbitrary N (> 1):
>
> As depicted in Algorithm 1, lines 7 and 8, during each iteration, each calibration network in the list of optimizing views connects to the same auxiliary network for training, aiming to minimize the prediction loss for distance to touchdown values. Moreover, as shown in lines 12-14 of Algorithm 1, views with loss values exceeding the minimum current loss of the calibration network by the defined threshold are selected. These views are then utilized for training in the next iteration. If all losses fall within a certain range, all calibration networks undergo training.
>
> 2. LSTM and Miss-Detection in Multi-Views:
>
> In this study, Long Short-Term Memory (LSTM) is employed to combine inputs from multiple views for distance prediction, effectively handling variations in input size. The primary purpose of this model is to address situations where miss-detection occurs in one of the views. When YOLO fails to detect an aircraft or exhibits low confidence, the detected output is discarded, avoiding its use in subsequent prediction steps. This leads to variations in the input size during the inference step.
>
> In cases of multiple aircraft in each view, even if the aircraft appearances are similar from a distance, their positions vary significantly based on their distance to touchdown. In these aircraft landing scenarios, all aircraft adhere to specific air traffic control procedures. Therefore, the aircraft from multiple views are matched based on their related positions on the screen and their estimated distance to touchdown from the auxiliary network.
>
> 3. Dataset:
> The simulated dataset featuring multiple aircraft will be made publicly available upon request. However, the dataset obtained from real airport environments is considered highly confidential and cannot be shared.

---

### Official Review · Reviewer_Z485 · 2023-10-25

**Soundness:** 2 fair
**Presentation:** 2 fair
**Contribution:** 1 poor
**Rating:** 3
**Confidence:** 4

**Summary:**

This work proposes a multi-view deep learning approach for distance-touchdown (DTD) estimation. Yolov7 is utilized here to detect the aircraft in image. Input vecotrs from different views are further combined in an LSTM model, resulting the estimated distance.

**Strengths:**

1. The experiments on simulated and real video data demonstrate that the proposed method can favorably extimate the distance of the aircraft.

**Weaknesses:**

1. This manuscript sounds more like a technique report instead of a research paper. The proposed approach simply utilize an off-the-shelf detection model and an LSTM network to train a distance estimation model.
2. The authors are encourged to evaluate the performance of baselines with different detection models and network structures.

**Questions:**

When there are more than one aircrafts in the frame, how do you associate the aircrafts across different views?

---

> ### Author Response · Authors · 2023-11-22
> **Discussion on the key contributions of the papers based on reviewer comments**
>
> Thank you for your constructive comments and feedback.
>
> 1. Calibration Network, Auxiliary Network, and Their Training:
> The primary contribution of this work lies in proposing a framework that optimally utilizes multiple view camera streams in airport digital towers to estimate the distance to touchdown of landing aircraft. The approach is grounded in ensemble learning, where each view is processed independently, and their information is subsequently transformed into a shared space for ensemble prediction. The proposed framework is designed to provide flexible adaptation for different digital tower configurations. Depending on the available airport data and its corresponding challenges, the model for aircraft detection can be customized to maximize the system's performance.
>
> Besides, the calibration network plays a pivotal role in handling multiple cameras by converting input features from individual views into a shared feature space. Furthermore, an auxiliary network is shared and employed by all calibration networks to train their parameters.  During training, the calibration network is jointly trained with the auxiliary head, but it undergoes separate training sessions with other instances of the calibration network. Therefore after the training process, the weights of the calibration network become associated with the specific camera and its configuration, while the weight of the auxiliary head is general for every camera. The output of the calibration networks is considered as the shared feature space.
>
> Furthermore, the current choice of the number of views (e.g., 2) aligns with the current operational constraints of digital towers. However, Algorithm 1 is designed to accommodate arbitrary N (> 1):
>
> As depicted in Algorithm 1, lines 7 and 8, during each iteration, each calibration network in the list of optimizing views connects to the same auxiliary network for training, aiming to minimize the prediction loss for distance to touchdown values. Moreover, as shown in lines 12-14 of Algorithm 1, views with loss values exceeding the minimum current loss of the calibration network by the defined threshold are selected. These views are then utilized for training in the subsequent iteration until their loss value is within a said threshold. If all losses fall within a certain range, all calibration networks undergo training.
>
> 2. LSTM and Miss-Detection in Multi-Views:
>
> In this study, Long Short-Term Memory (LSTM) is employed to combine inputs from multiple views for distance prediction, effectively handling variations in input size. The primary purpose of this model is to address situations where miss-detection occurs in one of the views. When YOLO fails to detect an aircraft or exhibits low confidence, the detected output is discarded, avoiding its use in subsequent prediction steps. This leads to variations in the input size during the inference step.
>
> In cases of multiple aircraft in each view, even if the aircraft appearances are similar from a distance, their positions vary significantly based on their distance to touchdown. In these aircraft landing scenarios, all aircraft adhere to specific air traffic control procedures. Therefore, the aircraft from multiple views are matched based on their related positions on the screen and their estimated distance to touchdown from the auxiliary network.
>
> 3. Dataset:
> The simulated dataset featuring multiple aircraft will be made publicly available upon request. However, the dataset obtained from real airport environments is considered highly confidential and cannot be shared.

---

### Official Review · Reviewer_kvrA · 2023-10-31

**Soundness:** 1 poor
**Presentation:** 2 fair
**Contribution:** 1 poor
**Rating:** 1
**Confidence:** 4

**Summary:**

The paper proposes a deep learning solution that uses images captured by cameras installed at airport runways to monitor aircraft traffic around an airport to estimate distance-to-touchdown for incoming (i.e., landing) aircrafts.  Distance-to-touchdown is an important piece of information that is used by airtraffic controllers to manage the air traffic.  The work proposed here develops a key enabling technology for the future digital (air taffic control) towers.  The proposed method is able to integrate information captured by multiple cameras in order to carry out the task at hand.  Each camera feed is processed independently to detect and segment the incoming aircrafts.  Camera network layers process features computed at each camera and the result is sent to an LSTM+inference network.  An auxiliary regression task is used to improve training.  The work is evalauted on both synthetic data, rendered using the popular X-Plane 11 flight simulator and on real data collected at the Singapore Changi airport.

**Strengths:**

The paper tackles an important problem in aircraft traffic management and control.  Clearly, vision-based automated schemes for detecting, identifying, and tracking air traffic in and around an airport is of immense value.  The paper cogently argues the need for such a system.  The paper also makes a clever use of synthetic data to train and evaluate the distance-to-touchdown estimation model.  The paper also makes use of TensorRT engine to speed up inference.  This is important due to the real-time nature of the task that the paper wants to solve.

**Weaknesses:**

The work as presented suffers from a number of weaknesses.

First off, majority of training and evaluation takes place in a setting that uses only two cameras.  This is unsatisfactory given that the multi-view analysis is one of the central claims of this work.

It is not immediately obvious how the architecture depicted in Figure 1 manages to integrate the information from multiple cameras.  It seems that the "calibration network" is tasked with transforming the features captured by multiple cameras into a shared space where these can be reasoned with jointly.  I feel that we need a lot more discussion around this "calibration network" and how it helps integrate information from multiple cameras.

Part of the "inference" network contains an LSTM.  It is not clear to me if LSTM is needed to deal with a single frame from multiple cameras or if LSTM is needed to process video feeds.  It appears to me that temporal information may be helpful in regularizing the distance-to-touchdown estimates.  Does the system uses temporal information?

What role does auxiliary network play?  And more importantly how does it play the said role?  What is a reversed network?

The overall scheme seems rather ad hoc.  YOLO is used as an object detector here.  What if it fails to record a plane?  What if planes are mis-labelled in multiple views?  At a distance most planes look similar!

Some of the discussion around results raises questions.  On page 8, why does the system perform better in low-light conditions.  This is very counter-intuitive.  This is a safety critical application, so the bar of scientific rigour is very high.  It is not sufficient that the proposed system achieves good results.  It is also important that we understand the limits of this system.  We should be able to explain good (or bad) results.  Perhaps an ablative study will help explain the roles played by individual components of the system.

**Questions:**

1. What happens if the N_views are set to more than 2 in Algorithm 1?
2. Why max_epoches are set to 225K in Algorithm 1?  This seems an arbitrary number.
3. Why does the system perform better in low-light conditions?    Do we know why?
4. How does the system deal with the object association problem in multiple images?
5. Not sure I can understand the second sentence in the Conclusions section.  It refers to stochastic number of input videos ...  What does it actually mean?
6. The paper refers to TensortRT?  Do you mean TensorRT?
7. What is a reversed network?
8. Can you provide some details of the auxiliary regression network?
9. Can you provide some details of the calibration network?  It may be useful to provide an ablative study on this, since it plays a central role in integrating information from multiple cameras.
10. What role does LSTM play?  Is it used to combine information from multiple cameras?  Or it integrates information over time to provide better distance-to-touchdown estimates.

---

> ### Author Response · Authors · 2023-11-22
> **Discussion on the key contributions of the papers based on reviewer comments**
>
> Thank you for your constructive comments and feedback.
>
> 1. Calibration Network, Auxiliary Network, and Their Training:
> The primary contribution of this work lies in proposing a framework that optimally utilizes multiple view camera streams in airport digital towers to estimate the distance to touchdown of landing aircraft. The approach is grounded in ensemble learning, where each view is processed independently, and their information is subsequently transformed into a shared space for ensemble prediction.
>
> And yes, the calibration network plays a pivotal role in handling multiple cameras by converting input features from individual views into a shared feature space. Furthermore, an auxiliary network is shared and employed by all calibration networks to train their parameters.
>
> Furthermore, the current choice of the number of views (e.g., 2) aligns with the current operational constraints of digital towers. However, Algorithm 1 is designed to accommodate arbitrary N (> 1):
>
> As depicted in Algorithm 1, lines 7 and 8, during each iteration, each calibration network in the list of optimizing views connects to the same auxiliary network for training, aiming to minimize the prediction loss for distance to touchdown values.
>
> Moreover, as shown in lines 12-14 of Algorithm 1, views with loss values exceeding the minimum current loss of the calibration network by the defined threshold are selected. These views are then utilized for training in the next iteration. If all losses fall within a certain range, all calibration networks undergo training.
>
> 2. LSTM and Miss-Detection in Multi-Views:
>
> In this study, Long Short-Term Memory (LSTM) is employed to combine inputs from multiple views for distance prediction, effectively handling variations in input size. The primary purpose of this model is to address situations where miss-detection occurs in one of the views. When YOLO fails to detect an aircraft or exhibits low confidence, the detected output is discarded, avoiding its use in subsequent prediction steps. This leads to variations in the input size during the inference step.
>
> In cases of multiple aircraft in each view, even if the aircraft appearances are similar from a distance, their positions vary significantly based on their distance to touchdown. In these aircraft landing scenarios, all aircraft adhere to specific air traffic control procedures.
>
> 3. Dataset:
> The simulated dataset featuring multiple aircraft will be made publicly available upon request. However, the dataset obtained from real airport environments is considered highly confidential and cannot be shared.

---

> > ### Author Response · Authors · 2023-11-22
> > **Response to reviewer's questions**
> >
> > 1. What happens if the N_views are set to more than 2 in Algorithm 1?
> > > Algorithm 1 is intended for use when N_views > 0, hence it should work as intended. In scenarios where multiple views exhibit loss values surpassing the threshold, the adaptive training algorithm will prioritize the view with the highest loss.
> > 2. Why max_epoches are set to 225K in Algorithm 1? This seems an arbitrary number.
> > > Our observation indicates that, typically, the model achieves satisfactory performance without overfitting at 225k epochs. Nevertheless, it's essential to note that the optimal value of this epoch number may vary based on the specific characteristics of the data under consideration.
> > 3. Why does the system perform better in low-light conditions? Do we know why?
> > > We suspect that this phenomenon could be attributed to the contrast between the aircraft and its surroundings. The aircraft used in our experiments is a white Boeing 737, and this contrast may become more pronounced in darker skies, potentially contributing to the observed effects.
> > 4. How does the system deal with the object association problem in multiple images?
> > > Currently, we employ a rule-based approach for associating aircraft across different views. We consider this method adequate, given the typically well-organized nature of air traffic.
> > 5. Not sure I can understand the second sentence in the Conclusions section. It refers to stochastic number of input videos ... What does it actually mean?
> > > This implies that the quantity of detections in video input may fluctuate from one timestep to another. Such variations can arise from either missed detections or instances where detection is excluded due to a confidence score falling below a specified threshold.
> > 6. We appreciate the clarification. We did intend to refer to TensorRT.
> >
> > 8. Can you provide some details of the auxiliary regression network?
> > > The auxiliary network is a shallow network specifically crafted to extract information, such as distance-to-touchdown, from a shared feature space derived from individual views. A singular instance of this network is employed for all cameras. The rationale behind this design is based on the notion that if a shallow network consistently predicts the distance-to-touchdown from the shared feature space of different cameras, then that space must have effectively encoded the pertinent information.
> > 9. Can you provide some details of the calibration network? It may be useful to provide an ablative study on this, since it plays a central role in integrating information from multiple cameras.
> > > The calibration network is formulated to encode the distance-to-touchdown within a feature space derived from the aircraft bounding box, while simultaneously considering the properties and configuration of the camera. Consequently, the model acquires the ability to correlate the size and position of the bounding box in the frame with the aircraft distance. Recognizing that diverse camera configurations may influence this association, we employ N calibration networks for N different cameras, each specifically customized to the characteristics of the individual camera.
> > 10. What role does LSTM play? Is it used to combine information from multiple cameras? Or it integrates information over time to provide better distance-to-touchdown estimates.
> > > The LSTM plays a crucial role in this architecture by treating the N shared feature spaces from N camera views as a sequence with a length of N. In essence, the LSTM facilitates the integration of information across the camera views, allowing the model to generate a sensible prediction, albeit with reduced accuracy, in cases where missed detections occur.

---

### Meta-Review · Area_Chair_7yex · 2023-12-06

**Metareview:**

This work presents a method for the estimation of the time to land of aircraft in an air monitoring situation, which is based on detections with the YOLO object detector. The paper had received three highly critical reviews, with reviewers raising issues on novelty and lack of methodological contributions, justification of the key design choices, positioning with respect to prior work, and comparisons with competing methods.

The answers provided by the authors could not clear up this issues, and the reviewers and the AC judge that this work is not suited for publication at ICLR 2024.

**Justification For Why Not Higher Score:**

This was a clear decision

**Justification For Why Not Lower Score:**

This was a clear decision

---

### Decision · Program_Chairs · 2024-01-16

Reject